# Synergy of Body Composition, Exercise Oncology, and Pharmacokinetics: A Narrative Review of Personalizing Paclitaxel Treatment for Breast Cancer

**DOI:** 10.3390/cancers17081271

**Published:** 2025-04-09

**Authors:** Nele Adriaenssens, Stephanie C. M. Wuyts, Stephane Steurbaut, Pieter-Jan De Sutter, An Vermeulen, Amy de Haar-Holleman, David Beckwée, Steven Provyn, Sofie Vande Casteele, Jinyu Zhou, Katrien Lanckmans, Jan Van Bocxlaer, Len De Nys

**Affiliations:** 1Rehabilitation Research, Vrije Universiteit Brussel (VUB), Laarbeeklaan 121, 1090 Brussels, Belgiumjinyu.zhou@vub.be (J.Z.); len.de.nys@vub.be (L.D.N.); 2Medical Oncology Department, Universitair Ziekenhuis Brussel (UZ Brussel), Laarbeeklaan 101, 1090 Brussels, Belgium; 3Pharmacy Department, Universitair Ziekenhuis Brussel (UZ Brussel), Laarbeeklaan 101, 1090 Brussels, Belgiumstephane.steurbaut@uzbrussel.be (S.S.); 4Research Centre for Digital Medicine, Vrije Universiteit Brussel (VUB), Laarbeeklaan 103, 1090 Brussels, Belgium; 5Vitality Research Group, Vrije Universiteit Brussel (VUB), Laarbeeklaan 103, 1090 Brussels, Belgium; 6Department of Bioanalysis, Faculty of Pharmaceutical Sciences, Universiteit Gent, Ottergemsesteenweg 460, 9000 Gent, Belgiumsofie.vandecasteele@ugent.be (S.V.C.);; 7Translational Oncology Research Center, Vrije Universiteit Brussel (VUB), Laarbeeklaan 103, 1090 Brussels, Belgium; 8Human Physiology and Sports Physiotherapy, Vrije Universiteit Brussel (VUB), Pleinlaan 2, 1050 Brussels, Belgium; 9Clinical Biology Department, Universitair Ziekenhuis Brussel (UZ Brussel), Laarbeeklaan 101, 1090 Brussels, Belgium; katrien.lanckmans@uzbrussel.be

**Keywords:** body composition, paclitaxel, chemotherapy toxicity, pharmacokinetics, physical activity, exercise oncology

## Abstract

Paclitaxel is a chemotherapy drug widely used for breast cancer, but serious side effects, such as nerve damage and low white blood cell counts, often limit its effectiveness. Currently, chemotherapy doses are based on body surface area, but this method does not consider important individual factors like muscle and fat levels, which can affect how the drug is processed. This review explores how body composition and physical activity influence paclitaxel’s effects and side effects. Regular exercise may improve treatment outcomes by supporting muscle health and reducing toxicity risks. By integrating these factors into chemotherapy dosing strategies, we can move toward a more personalized approach, ensuring that each patient receives the most effective and safest treatment. This research highlights the need for further studies to refine chemotherapy dosing, potentially leading to better treatment experiences and outcomes for breast cancer patients.

## 1. Introduction

Breast cancer (BC) is the most common cancer among women worldwide, accounting for 11.7% of all cancer cases, with an estimated 2.26 million new diagnoses globally in 2020 [1,2,3]. An estimated 355,000 new cases of BC were diagnosed in the European Union (EU) in 2020, representing approximately 13.3% of all cancer cases in the region. This marks BC as a significant health concern, as it also accounted for 7.3% of cancer-related deaths within the EU in 2020 [4]. Over recent decades, the survival rates and life expectancies of BC patients have significantly improved, mainly due to improvements in early detection through screening and advancements in chemotherapy and targeted therapies [4].

Depending on the stage and type of BC, treatment strategies primarily include local therapies, such as surgery with or without radiation, and systemic therapies, including chemotherapy, hormone therapy, targeted therapies, and/or immunotherapy According to major prospective clinical trials, several standard chemotherapy regimens, including anthracyclines and taxanes, are available for BC treatment [5,6]. Taxane-based regimens, such as paclitaxel (PTX), are among the most effective and commonly used systemic therapies for both early- and late-stage BC [7]. However, the toxicity associated with chemotherapy remains a significant challenge in the treatment of BC [8].

In this narrative review, we explore how interindividual differences in body composition and physical activity levels may influence paclitaxel pharmacokinetics and the risk of chemotherapy-induced toxicities in breast cancer patients. We synthesize current evidence on the relationships between skeletal muscle mass, adipose tissue distribution, physical activity, and paclitaxel metabolism and clearance. Special attention is given to how unfavorable body composition—such as sarcopenia or sarcopenic obesity—can increase the risk of dose-limiting toxicities and how physical activity may mitigate these effects by positively influencing body composition and metabolic function (Figure 1). Finally, we highlight the emerging role of exercise oncology and its potential to contribute to more personalized, safe, and effective chemotherapy strategies.

## 2. Paclitaxel in the Treatment of Breast Cancer

### 2.1. Toxicities

Paclitaxel induces cancer cell death through microtubule stabilization, leading to a mitotic arrest [9]. However, its mechanism of action can also affect healthy dividing cells, resulting in side effects. Patients with BC undergoing PTX therapy may experience a variety of toxicities, including neurological, hematological, gastrointestinal, and cardiac adverse effects, as well as hypersensitivity reactions [8,10]. The severity and presentation of these side effects can vary among patients and are often influenced by the treatment schedule and dosing protocols [10].

The toxicities associated with PTX continue to limit the effectiveness of treatment regimens based on this drug [10]. For example, while pretreatment with a standard regimen that includes a corticosteroid (such as dexamethasone) and an H1 receptor blocker can help reduce the risk of hypersensitivity reactions before PTX infusions, minor hypersensitivity reactions still occur in a significant number of patients [10,11]. Importantly, a small but critical percentage of individuals still experience life-threatening reactions [12].

The primary hematologic toxicity associated with PTX is neutropenia [13], which can be managed with the administration of granulocyte colony-stimulating factor. This leaves neurotoxicity, particularly chemotherapy-induced peripheral neuropathy (CIPN), as the main dose-limiting toxicity (DLT) [14]. CIPN, which can develop either early or late after PTX administration, affects the majority of patients undergoing treatment [15]. Clinically, it is characterized by numbness and paresthesia in a glove-and-stocking distribution, resulting from the accumulation of PTX in the dorsal root ganglia [16,17]. These DLTs can result in treatment delays, dose reductions, or even premature termination of therapy [18]. Additionally, they significantly impact patients’ health-related quality of life (HRQoL) and functional status in both the short and long term [19].

### 2.2. PTX Dose Intensity

Relative dose intensity (RDI), the ratio of the actual delivered dose intensity (mg/m^2^ per week) to the planned dose intensity for a chemotherapy regimen, is a key measure that reflects dose delays or reductions during treatment [20,21,22]; however, its application must be interpreted in the context of evolving evidence highlighting the limitations of BSA-based dosing (see below in Section 3). While an RDI below 85% is considered a clinically significant reduction compared to the standard therapy, these thresholds are typically derived from standard dosing models that do not consider individual patient characteristics, such as body composition. This decreased RDI is associated with worse outcomes, including reduced survival rates in advanced BC. This association has been demonstrated in both randomized clinical trials and retrospective observational studies [20,23,24]. Maintaining an adequate RDI is particularly important for BC patients undergoing PTX therapy, as timely administration of the planned dosage is crucial for achieving optimal prognosis. In most cancers, there is a plateau in the dose–response curve for cytotoxic chemotherapy. This means that increasing the dose towards an asymptotic threshold leads to increased toxicity without providing additional antitumor effects [25]. Therefore, it is essential for patients undergoing chemotherapy to achieve a therapeutic response while minimizing toxicity, ensuring that their drug exposure remains at acceptable levels.

Studies have explored this dose–response relationship, specifically for PTX; however, evidence regarding PTX’s therapeutic and toxic ranges is limited to certain dosing schedules [26]. Key toxicity parameters, such as CIPN, have been associated with elevated maximal PTX plasma concentrations (Cmax) or prolonged exposure durations above the threshold of 0.05 µM [27,28,29] It is crucial to monitor and manage these toxicities to ensure an optimal RDI for patients with BC throughout their chemotherapy treatment.

### 2.3. Challenges Regarding Body Surface Area Dosing of PTX

Body surface area (BSA) has long been the standard metric for determining chemotherapeutic drug dosage, as described in two reviews from 2016 [30,31]. The normalization of chemotherapy dosing was initially recommended because of the proposed relationship with other physiologic parameters, including resting energy expenditure, plasma volume, and cardiac output [32]. Various BSA formulae have been developed to simplify this process [31]. All recognized formulae are based on easily identifiable patient body variables such as height and weight and in some cases, age and sex [31].

However, the use of BSA as a basis for chemotherapy dosing, including PTX in BC, has long been debated due to its poor correlation with physiological parameters influencing pharmacokinetics and pharmacodynamics (PK/PD), such as differences in skeletal muscle mass (SMM), adipose tissue (AT), and metabolic activity [31]. As Miller et al. (2004) demonstrated two decades ago, BSA is proportional to blood volume, but does not accurately reflect an individual’s ability to metabolize or excrete cytotoxic drugs [33]. Their study found no significant association between BSA and PTX-induced toxicities, such as leukopenia or neutropenia, suggesting that BSA is not a reliable predictor of chemotherapy toxicity.

## 3. Role of Body Composition and Physical Activity in Breast Cancer Treatment

### 3.1. Body Composition and Chemotherapy Dosing

Patients exhibit significant biological heterogeneity, which can affect the desired patient outcomes. Even individuals with similar or identical body weight, BSA, or body mass index (BMI) can show considerable variations in the amount and distribution of AT and lean body mass, including SMM and extracellular fluid [34]. Body composition can influence the pharmacokinetics of chemotherapy in several ways. Therefore, it has been suggested that considering body composition parameters may be more accurate than BSA for calculating the appropriate chemotherapy dosage [35].

Changes in body composition, such as reduced SMM, decreased muscle quality, and increased AT, significantly contribute to DLTs across various cancer treatments, irrespective of cancer type, age, sex, BMI, or physical function [35,36,37,38]. While there is a correlation between BMI and both SMM and AT, there is considerable variability in these measurements for any given BMI or BSA value, particularly in women with BC [34,39]. Substantial evidence supports the role of SMM and AT as predictive factors for the occurrence of DLT during chemotherapy [40,41,42,43]. Additionally, studies have shown that toxicities from anthracyclines and taxanes, such as PTX, both commonly used in BC treatment, are more closely related to SMM than to BSA [44]. In patients with metastatic BC, sarcopenia—which is defined as a reduction in SMM—has been associated with an increased risk of chemotherapy toxicities and a shorter median time to tumor progression, irrespective of overall BMI [26,43,44,45]. This association is particularly significant in patients with sarcopenic obesity, characterized by low SMM and high AT. Sarcopenic obesity, along with conditions like myosteatosis (fat infiltration in muscle), can lead to changes in drug distribution, metabolism, and exposure. This may lead to over (or under) dosing, increased systemic inflammation, and decreased physical resilience [36,43,46,47]. If chemotherapy dosing is based on BSA, patients with sarcopenia may receive disproportionately high doses relative to their metabolically active SMM, further increasing toxicity risks [38,48]. Similar findings have been reported in other cancer types, where patients with low SMM are at a higher risk of experiencing treatment-related toxicities [40,49,50,51,52].

Additionally, the body composition of patients with metastatic BC often differs from that of early-stage BC patients due to cancer-related fatigue, sarcopenia and/or cachexia, and treatment-related side effects such as bone demineralization (osteopenia/osteoporosis), reduced muscular strength [53], decreased aerobic capacity [54], and weight gain [55,56]. These observations highlight the need to better understand the role of body composition in chemotherapy tolerance among women with BC [34].

Advances in standard imaging techniques for body composition assessment, such as dual-energy X-ray absorptiometry (DXA) and single-slice computed tomography (CT) scanning, have led to numerous studies exploring the associations between body composition and adverse outcomes in oncology [18,57,58]. Recently, artificial intelligence has been utilized to extract body composition metrics from medical images, providing new opportunities for facilitating personalized treatment approaches [59]. Given the widespread use of CT imaging in cancer diagnosis and monitoring, body composition can be analyzed, without imposing an additional burden on patients, using available diagnostic images [60].

### 3.2. Body Composition, Paclitaxel Pharmacokinetics and Toxicities

In contrast to the numerous studies that have investigated body composition and cancer outcomes, only a limited number of trials have explored the correlation between body composition and small molecule drug PK [35]. Each drug exhibits intrinsic properties that contribute to variability in its distribution throughout the body. Hydrophilic drugs primarily distribute into lean tissue, while lipophilic drugs accumulate in the AT, affecting their volume of distribution (Vd) and elimination rate from the body [61]. Patients with sarcopenic obesity exhibit a reduced Vd for hydrophilic drugs and an increased Vd for lipophilic drugs. These alterations in drug distribution are not adequately accounted for by BSA-based dosing [38]. Conversely, the PK of large-molecule drugs, such as monoclonal antibodies (mAbs), are expected to be less influenced by body composition, as their elimination is receptor-mediated, and their distribution is primarily limited to the extracellular space [62]. A conceptual model of interactions between body composition and PTX pharmacokinetics is shown in Figure 2.

Several examples were identified for the chemotherapy drugs docetaxel, epirubicin, and oxaliplatin. A recent study confirms that overweight and obese patients receiving adjuvant chemotherapy based on docetaxel, another taxane, exhibit worse disease-free survival and overall survival rates, as well as a higher risk of distant metastases, compared to those of lean patients [61]. In contrast, no significant differences in outcomes were observed across BMI categories for patients treated with non-docetaxel-based chemotherapy. These findings may be attributed to docetaxel’s lipophilic properties, leading to an increased Vd and reduced efficacy in patients with a higher BMI. Although the study discusses BMI rather than body composition, the authors suggested that the risk–benefit ratio of using taxanes in BC should be reassessed based on the patients’ body composition [61]. Research conducted by Prado et al. (2011) demonstrated that variability in SMM was linked to toxicities and differences in the drug clearance of epirubicin in cancer patients, while BSA was not a reliable predictor [63]. More recent research on the PK of oxaliplatin in older adults with gastrointestinal malignancies indicated that patients with low SMM and high total AT exhibited the lowest Vd and drug clearance, the highest maximal drug concentrations, and a significantly increased risk of severe chemotherapy toxicities [64].

The hydrophobic nature of PTX leads to substantial accumulation in AT, high protein binding, and delayed clearance, contributing to its complex PK profile and toxicity risks [10,64]. A decrease in SMM is believed to influence PK by reducing the Vd and affecting protein binding, drug metabolism, and clearance [35]. SMM has been reported to influence the PK of certain chemotherapeutic agents, including PTX [27,35,36,63,65,66]. Specifically, lower SMM, such as in patients with sarcopenia, has been associated with an increase in PTX’s maximal concentration (Cmax), which likely explains the increased risk of CIPN in these patients. PK modeling simulations suggest that extending the infusion of PTX in patients with the lowest SMM may reduce CIPN while maintaining therapeutic efficacy [27]. Furthermore, the active compound and that of the excipient(s) may affect drug exposure. In a study by Smorenburg et al. (2003), older adult patients with BC displayed up to a 50% increase in PTX exposure with age [67]. This increase was attributed to a significantly altered disposition of the formulation vehicle, Cremophor EL, which was markedly decreased in the older patient group. The potential association between changes in body composition, and this decrease in PTX total body clearance with age remains to be clarified [67]. Conversely, in more extensive cohort studies, older age was not identified as an independent risk factor leading to clinically relevant changes in PTX PK [68,69], suggesting that other variables, such as body composition, may be relevant.

Similarly, obesity may alter drug distribution and affect PK by decreasing the plasma concentrations of lipophilic drugs but increasing their storage in adipose tissue. For PTX, Sparreboom et al. found a 46.6% increase in Vd and a 20.4% increase in clearance in obese patients (BMI ≥ 30 kg/m^2^) compared to the results for lean patients (BMI < 25 kg/m^2^) [70]. Since PTX is strongly bound to plasma proteins (e.g., albumin), which are found to be reduced in obese patients [71], this may lead to an increase in the free fraction of PTX, which in turn may increase its pharmacological effect, but also the risk of toxicity. PTX is metabolized in the liver by cytochrome P450 enzymes, primarily CYP3A4 and CYP2C8. Obese individuals may exhibit decreased drug metabolism, as individuals with nonalcoholic fatty liver disease can show impaired hepatic drug-metabolizing enzyme activity, resulting in prolonged PTX exposure and increased risk of adverse reactions. Although PTX is primarily metabolized by the liver and excreted in the bile, there is some renal excretion. Therefore, impaired renal function, as observed in some obese patients [72], can also adversely affect the elimination of this drug and its metabolites. In addition, women have been reported to experience greater toxicity than men at equivalent doses of PTX. This may be due to gender-related differences in body composition, such as women tending to carry more body fat than men. The increased susceptibility of women to the adverse effects of PTX has also recently been linked to differences in the functioning of the immune system, although the underlying mechanism remains poorly understood [73,74]. Therefore, it is important to be aware of interpatient variability and to consider differences in body composition when prescribing PTX.

More studies with adequate sample sizes and comprehensive pharmacologic endpoints are needed to examine the effects of body composition on chemotherapy PK, particularly for PTX [32]. Such studies will help validate preliminary findings and support the development of dosing strategies tailored to individual body compositions.

## 4. Role of Physical Activity and Exercise in Breast Cancer Treatment

In addition to body composition, physical activity (PA), defined as any movement resulting in energy expenditure, including leisure-time activities, and exercise, characterized as planned and structured PA aimed at improving physical capacity or physical fitness, is increasingly recognized as a crucial intervention for patients with BC during and after treatment [75,76]. Engaging in low-to-moderate levels of PA is well known to significantly reduce mortality in patients with various malignancies [77], offering greater health benefits compared to inactivity [78]. Evidence indicates that recreational PA can lower BC mortality risk, enhance physiological and immune functions [78,79], lower stress, and improve sleep disturbances and overall HRQoL [80]. Further, PA helps mitigate cancer- and treatment-related side effects, such as decreased muscular strength, weight gain, and bone demineralization, leading to improved body composition and better clinical outcomes in patients with BC [56,75,79,81,82].

The clinical impact of exercise throughout all phases of cancer care—ranging from prevention to advanced stages and end-of-life care—is well-documented [83,84]. Therefore, physiotherapists play a pivotal role in the interdisciplinary management of cancer, contributing therapies beyond exercise alone [85]. Exercise oncology, a relatively young therapy option discipline, emerged in the 1980s, primarily led by nurses. Early research established that exercise is feasible, safe, acceptable, and effective for individuals living with or beyond cancer [54,86,87,88].

In 2003, the first exercise guidelines for cancer were introduced [89]. In 2007, the Physical Activity Cancer Control framework was established to delineate the different phases of cancer care [90] This framework informed the first roundtable on exercise guidelines in oncology organized by the American College of Sports Medicine in 2010 [91], with subsequent updates and expansions in 2019 [92,93]. In 2022, the American Society of Clinical Oncology (ASCO) published guidelines regarding exercise during cancer treatment [94], as did the Dutch Royal Society for Physical Therapy (KNGF); these were later translated into English in 2023, and are now widely used in clinical practice [95].

Despite the advancements in the field of exercise oncology, two critical gaps persist. First, current guidelines predominantly emphasize rehabilitation after active medical treatment, such as chemotherapy, rather than during treatment. Second, they remain general in scope, primarily recommending PA levels similar to those for healthy populations (i.e., 150 min of moderate-intensity exercise per week or 10,000 steps per day, as per WHO guidelines) [96]. While the long-term effects of exercise on cancer-related side effects are well-documented in these guidelines, the acute and chronic physiological responses to exercise during chemotherapy remain underexplored. Positive changes in the tumor microenvironment might be driven by accumulative effects of repeated acute exercise responses [97,98,99]. These acute but short-lasting changes that exceed the adaptive responses produced by prolonged training might show the potential to provide immediate physiological benefits and in turn, improve DLTs [99].

Addressing these gaps is essential for optimizing exercise interventions tailored to the unique needs of patients undergoing active cancer treatment. Previous studies even suggest that exercise programs can improve DLTs like CIPN in patients undergoing taxane-, platinum-, or vinca alkaloid-based chemotherapy, as concluded in the review by Wirtz et al. (2018) [81] and confirmed in the more recent review by Tanay et al. (2023) [100]. Exercise also has the potential to prevent the onset and slow the progression of CIPN, particularly in patients with BC [81,101]. This reduction in side effects may be related to the multiple PK-related effects of exercise on chemotherapy that warrant further investigation. Further, exercise PK [102,103] and exercise oncology [104] are two young and promising research fields that play an important role in chemotherapy-induced DLTs.

Despite the clear benefits of exercise and PA in managing BC-related side effects, motivating women with BC to engage in regular PA remains a considerable challenge [105]. Post-treatment fatigue and pain related to chemotherapy and other treatments, along with other side effects, often discourage participation in PA [105]. Various studies have demonstrated that interventions such as cognitive behavioral strategies show added value in increasing PA motivation and adherence in women with BC by implementing approaches such as behavior change techniques [106,107,108] and motivational interviewing [109,110]. Additionally, patients with BC often reduce their PA during therapy [111] and tend to remain physically inactive after treatment [112].

## 5. Exercise-Mediated Improvements in Body Composition and Their Impact on Paclitaxel Tolerance

Chemotherapy often leads to detrimental changes in body composition, including the loss of SMM and increased adiposity, which are associated with poor physical function and adverse treatment outcomes [113,114]. Exercise interventions have shown promising potential in positively influencing body composition in cancer patients, which may be crucial for optimizing chemotherapy dosing and improving treatment outcomes [115].

Exercise, particularly supervised aerobic and resistance exercise, during chemotherapy enhances SMM, muscle strength, and endurance, while reducing AT accumulation and improving overall body composition in patients with BC. This is supported by the meta-analysis of Li et al. (2024) [116] and other recent publications [113,114,117,118,119,120,121,122,123,124,125,126]. Resistance training has emerged as a particularly effective strategy for preserving and even restoring skeletal muscle quality [122,123]. Mijwel et al. (2019) reported that a 16-week exercise intervention during chemotherapy for BC patients significantly improved muscle strength and quality [127]. Additionally, Aires et al. (2024) emphasize the role of exercise in rebuilding skeletal muscle integrity in BC patients post-chemotherapy [117]. They suggest that targeted PA interventions could help mitigate sarcopenia and its associated risks. These functional improvements in muscle tissue may contribute to increased chemotherapy tolerance and reduced incidence of DLTs.

While mechanistic differences exist between aerobic and resistance training, both modalities improve treatment tolerance by positively modulating body composition. Resistance training consistently increases lean mass (e.g., +8.3 kg versus +2.7 kg with aerobic training) [128] and reduces intermuscular fat infiltration, particularly beneficial for preserving skeletal muscle during chemotherapy [92,128]. In contrast, aerobic exercise yields more significant reductions in overall fat mass and percentage of body fat (e.g., −1.8% versus −0.9% for resistance training) [128].

Current ASCO guidelines recommend combining aerobic and resistance exercises to maximize clinical benefits across multiple outcomes [94]. Resistance training is especially emphasized in patients at risk for sarcopenia or who are undergoing dose-dense chemotherapy regimens. However, direct comparisons between aerobic and resistance exercise in regards to PTX pharmacokinetics are currently lacking. Preclinical models suggest that combined exercise may enhance drug delivery through mechanisms such as vascular normalization, but these findings require clinical validation [129]. Therefore, until PTX-specific data become available, exercise prescriptions should be individualized based on patient capacity, safety, and tolerance, with a preference for multimodal programs incorporating aerobic and resistance components. In the context of PTX, SMM is a critical factor influencing treatment outcomes and toxicities. Research by Wopat et al. (2024) and Poltronieri et al. (2022) reveals that lower SMM is associated with higher rates of DLTs, including CIPN and gastrointestinal distress [119,120]. Exercise-based interventions may mitigate these toxicities by maintaining SMM and improving metabolic function [125,126]. The review by Li et al. (2023) [116] further demonstrates that exercise during chemotherapy not only reduces treatment-related weight gain, particularly visceral adiposity, but also enhances chemotherapy tolerance by promoting a healthier body composition profile [116].

Adipose tissue, particularly visceral and subcutaneous fat, plays a nuanced role in chemotherapy outcomes. Excess adiposity has been linked to systemic inflammation and altered drug pharmacokinetics, exacerbating chemotherapy toxicities [121]. Exercise can attenuate these effects by reducing visceral fat and improving the visceral-to-subcutaneous adipose tissue ratio, as highlighted in the studies by Poltronieri et al. (2022) and Godinho-Mota et al. (2021) [113,120]. These improvements are particularly relevant for patients undergoing PTX, in which systemic inflammation is a key driver of DLTs [98,117].

Additionally, the dose and type of exercise matter. An et al. (2020) [125] found that higher doses of resistance exercise led to greater gains in SMM and reductions in toxicities. Meanwhile, Bland et al. (2022) argued for the integration of individualized exercise prescriptions tailored to patient needs and their baseline PA [122]. Such targeted interventions, as evidenced by Altundag (2020), not only improve the skeletal muscle index but also enhance HRQoL and reduce fatigue during chemotherapy [124]. Furthermore, Kudiarasu et al. (2023) report that combining exercise with dietary interventions yields significant improvements in both SMM and AT, supporting a multimodal approach to optimizing body composition [118].

## 6. Pharmacokinetics in Breast Cancer Treatment

Emerging research highlights the potential of exercise to influence drug pharmacology [32,102,130]. Exercise may reduce chemotherapy-related toxicities both indirectly, through improved physiological resilience, and more directly via modulating systemic drug exposure. In the context of PTX, dose-limiting toxicities, such as peripheral neuropathy, have been linked to higher Cmax and prolonged exposure above the toxicity threshold (Tc > 0.05 μM) [27,131]. Exercise-induced improvements in skeletal muscle mass and reductions in adiposity affect PTX pharmacokinetics, potentially lowering Cmax and shortening Tc > 0.05 due to changes in drug distribution and metabolism [102,119,131]. Additionally, exercise may enhance hepatic and renal perfusion, improve metabolic enzyme activity, and reduce systemic inflammation—factors that can promote drug clearance [102,132,133]. While direct clinical evidence of reduced systemic exposure achieved through exercise remains limited, recent studies, such as the PABTOX study, have highlighted this mechanism as a promising possibility for reducing toxicities and improving chemotherapy adherence [90,134,135]. The review by McLaughlin et al. (2017) provides a detailed overview of the physiological changes associated with acute, subacute, and chronic exercise, along with the respective drug PK variables that are affected [102].

During acute exercise, physiological changes include increased cardiac output; enhanced blood flow to active skeletal muscles, skin, digestive organs, kidneys, liver, and other tissues; and a decreased glomerular filtration rate [136,137]. Chronic effects are typically seen in body composition changes, such as increases in SMM and decreases in AT, along with increased serum albumin concentrations and a reduction in inflammatory cytokines [102]. The possible alteration in drug PK achieved via exercise is likely mediated through its impact on physiological processes that regulate a drug’s absorption, distribution, metabolism, and excretion [32,137,138] (Table 1).

First, regarding absorption, exercising may accelerate the absorption of drugs, e.g., when administered by intramuscular, subcutaneous, and transdermal routes or by inhalation due to improved blood flow [136,139]. However, the expected influence of exercise on drug absorption remains uncertain due to numerous factors, such as the physicochemical and biochemical properties of drugs, as well as individual anatomical and physiological variations [102].

Second, exercise can significantly impact drug distribution [136]. Acute exercise enhances splanchnic and hepatic blood flow, increasing drug delivery rates to targeted receptor sites and facilitating drug absorption and equilibrium between the plasma and tissues [32,102]. Chronic training increases lean body mass, which is well-hydrated, while decreasing AT. On the other hand, reduced blood flow to AT and other inactive regions during exercise may slow the distribution of lipophilic drugs, such as PTX, that rely on these compartments for storage [139]. Exercise can also alter plasma protein levels. An increase in plasma protein concentration during exercise, often due to hemoconcentration from reduced plasma volume, can enhance drug binding and decrease the free, active drug fraction. For highly protein-bound drugs like PTX, this effect may further reduce the free drug concentration during or after intense PA [102]. Additionally, exercise can affect blood pH and core body temperature, leading to minimal changes in drug binding and altering the amount of the free drug available in the bloodstream [102]. The clinical relevance of these changes to the final PK of the chemotherapeutic drug remains to be established.

Third, exercise can influence drug metabolism by modulating physiological factors such as plasma volume, capillarization of skeletal and cardiac muscle, cytochrome P-450 enzyme activity, and mitochondrial density [32]. During exercise, blood flow is redirected from the liver to the muscles, which can reduce the clearance of drugs with a flow-dependent hepatic metabolism, potentially influencing plasma concentrations [139]. An important chronic response to exercise is a reduction in inflammatory cytokines in cancer patients undergoing chemotherapy. A study by Schauer et al. (2021) [98] investigated the effects of high-intensity and low-to-moderate intensity exercise on inflammatory markers in patients with BC [98]. Over six months, patients engaged in combined aerobic and resistance exercise during and after chemotherapy. The study found that, regardless of exercise intensity, levels of interleukin 6 (IL-6), IL-8, IL-10, and tumor necrosis factor alpha (TNF-α) increased post-treatment, but generally declined post-intervention [98]. Notably, high-intensity exercise resulted in a smaller increase in C-reactive protein (CRP) and TNF-α immediately post-treatment compared to the results for low-to-moderate intensity exercise, suggesting that high-intensity exercise may offer better protection against chemotherapy-related inflammation [98]. While cytokines like IL-6 increase during the subacute period following exercise, their preliminary downregulation of the CYP450 genes is eventually outweighed by a more chronic response, leading to potential increases in the CYP450 microsomal content [102]. As the CYP-enzymes CYP3A4 and CYP2C8 are important for PTX metabolism, changes in their activity may influence overall metabolism [140,141].

Finally, exercise may also affect the elimination of water-soluble and renally extracted drugs and metabolites by decreasing their clearance, which in turn, can increase their plasma concentrations [136,139] depending on the exercise intensity [102]. Additionally, exercise can change urine pH, influencing drug ionization and affecting its reabsorption or excretion [102]. In contrast, exercise can increase the clearance of certain lipid-soluble drugs and/or metabolites by enhancing bile production and secretion, boosting biliary excretion, and reducing intestinal reabsorption, ultimately lowering plasma concentrations [102,136].

Currently, there is a limited amount of evidence regarding the impact of exercise on the bioavailability of anticancer drugs, highlighting the necessity for further research [130]. However, existing in silico models, such as the physiologically based PK model published by Guo et al. (2024), offer potential for predicting of the PK behavior of chemotherapeutic drugs during exercise [137].

## 7. Future Directions

Future research and clinical practice regarding exercise oncology should prioritize integrating body composition data, PA levels, and exercise into chemotherapy dosing decisions, moving beyond traditional BSA-based methods. This approach, supported by emerging evidence, promises to enhance the precision and personalization of chemotherapy, particularly for drugs like PTX with complex pharmacokinetic profiles.

In 2024, the PABTOX trial (NCT06387901) was launched at the Vrije Universiteit Brussel, Brussels, Belgium, in collaboration with the Universitair Ziekenhuis Brussel and the University of Ghent. This two-year translational research project investigates the interactions between PTX, DLTs, body composition, physical activity, and various patient-specific factors. This study aims to develop a predictive PK–PD model, based on these interactions, for DLTs in women with BC treated with PTX, potentially refining chemotherapy dosing and reducing adverse effects in the future [134].

In addition to PA and low to moderate exercise, high-intensity interval training (HIIT) during chemotherapy represents a promising area for further exploration. Limited evidence suggests that HIIT may offer superior benefits in preserving SMM, improving fitness, and reducing AT compared to the results for traditional exercise [91,127]. Given the challenges of therapy adherence and motivation during chemotherapy, HIIT’s time-efficient nature could enhance patient engagement and treatment outcomes (e.g., NCT05786014, NCT04724499, NCT05913713). Future studies should explore the feasibility and efficacy of integrating HIIT into exercise oncology programs, focusing on its impact on body composition and DLT risks in breast cancer patients.

Moreover, the acute responses to exercise and their potential to directly influence DLTs remain underexplored. Understanding these acute responses could provide immediate physiological benefits and improve chemotherapy tolerance [97,98]. Positive changes in the tumor microenvironment driven by repeated acute exercise responses may explain observed benefits such as higher chemotherapy completion rates and improved aerobic capacity [99]. Future research should aim to elucidate these mechanisms and refine exercise prescriptions to maximize therapeutic benefits.

Additionally, integrating nutritional guidelines from the European Society for Clinical Nutrition and Metabolism (ESPEN) into the care of patients with BC during chemotherapy can significantly enhance the benefits of exercise. ESPEN recommends a daily caloric intake of 25–30 kcal/kg body weight and a protein intake of 1.2–1.5 g/kg body weight to support anabolic processes and maintain SMM [142,143]. This nutritional support is crucial for optimizing body composition, as adequate protein intake is essential for muscle repair and growth, particularly when combined with resistance and aerobic exercise [142,143]. By ensuring that patients meet these nutritional targets, healthcare providers can help maximize the effectiveness of exercise interventions, reduce the incidence of DLTs, and improve overall treatment outcomes. This multimodal approach, combining tailored exercise programs with precise nutritional support, offers a comprehensive strategy to enhance HRQoL and treatment efficacy for breast cancer patients undergoing chemotherapy. There are already several examples of (ongoing) studies combining exercise and nutritional interventions in cancer care, with promising results [135,144].

Clinically, low SMI may help identify high-risk patients who could benefit from extended infusion protocols, closer monitoring, or exercise interventions to improve muscle mass and treatment tolerance [27,122,123]. To facilitate this, oncologists could use routinely acquired CT or PET-CT scans to assess e.g., SMI at the L3 level, a suggested proxy for total muscle mass that needs to be further explored [43,59,145]. Automated tools using artificial intelligence can extract SMI from scans without added burden [59,60,145]. As imaging tools become standardized, integrating SMI into pharmacokinetic-informed dosing algorithms represents a promising step toward personalized chemotherapy. Beyond SMI alone, the ongoing PABTOX study (NCT06387901) mentioned above is developing a PK–PD model to integrate additional personalized metrics, i.e., SMM, AT, and PA, to improve early identification of patients at risk for dose-limiting toxicities and low relative dose intensity.

## 8. Conclusions

This review highlights the limitations of BSA-based dosing for PTX in patients with BC and emphasizes the need to consider individual variations in body composition when optimizing treatment strategies. Differences in body composition influence the risk of toxicities, underscoring the importance of more personalized approaches to chemotherapy dosing.

Exercise and PA have emerged as modifiable factors that could positively impact treatment outcomes by improving overall physical health and enhancing chemotherapy tolerance. However, the specific mechanisms by which exercise influences the PK of chemotherapeutic drugs, including PTX, remain underexplored. Future research is critical for filling this knowledge gap and developing strategies to effectively incorporate exercise into standard treatment protocols.

By integrating body composition assessments and structured exercise (and nutritional) interventions into PTX treatment for BC, therapies can be customized for more personalized and effective outcomes, as these strategies hold significant potential for reducing DLTs, improving PTX efficacy, and enhancing the HRQoL for BC patients.

As the fields of exercise oncology and personalized medicine continue to advance, exercise programs and body composition-based dosing strategies are expected to play an increasingly important role in BC care. Translational research is required to move beyond BSA-based dosing, exploring the potential value of body composition variables in PTX PK. In 2024, the PABTOX trial (Investigating Paclitaxel Toxicity in Breast Cancer: the Roles of Physical Activity and Body Composition—NCT06387901) started patient inclusion, aiming to provide the first preliminary evidence. Future research should focus on developing and validating these approaches to improve the efficacy and tolerability of PTX treatment for BC patients.

## Figures and Tables

**Figure 1 cancers-17-01271-f001:**
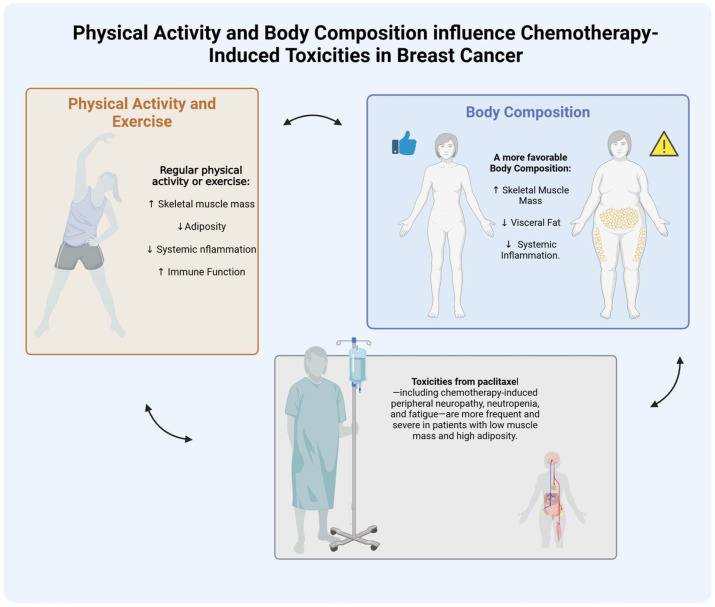
Illustration showing how physical activity and body composition influence chemotherapy-induced toxicities in breast cancer patients. A more favorable body composition—characterized by higher skeletal muscle mass, lower visceral fat, and reduced systemic inflammation—is associated with a lower risk of chemotherapy-induced toxicities such as peripheral neuropathy, neutropenia, and fatigue.In contrast, patients with low muscle mass and high adiposity are more susceptible to treatment-related toxicities due to altered drug metabolism, distribution, and increased systemic inflammation.Curved arrows represent cyclical and interdependent relationships between physical activity, body composition, and treatment tolerance. Improving physical activity can positively shift body composition and reduce toxicities, which may in turn support continued or enhanced engagement in exercise. *Created in BioRender, https://BioRender.com*.

**Figure 2 cancers-17-01271-f002:**
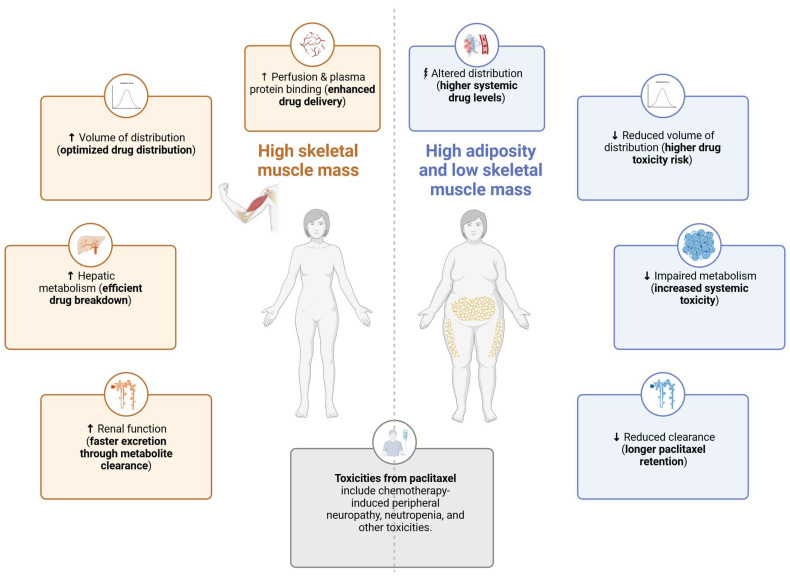
Conceptual model of interactions between body composition and PTX pharmacokinetics. Upward arrows (↑) indicate increases or enhancements in physiological or pharmacokinetic processes (e.g., increased hepatic metabolism or renal clearance), which are generally associated with improved drug handling and reduced toxicity risk.Downward arrows (↓) indicate reduced function or capacity, such as decreased volume of distribution or impaired metabolism, typically associated with higher systemic drug levels and increased toxicity risk.Lightning bolts (⚡) indicate altered or dysregulated processes, such as disrupted drug distribution, which may contribute to unpredictable paclitaxel exposure. *Created in BioRender, https://BioRender.com*.

**Table 1 cancers-17-01271-t001:** Potential mechanisms by which exercise and body composition modulate paclitaxel pharmacokinetics.

Pharmacokinetic Process	Exercise Effect	Proposed Mechanism	Relationship to Body Composition	Clinical Implications
**Absorption**	Minimal direct effect (for IV drugs); potential changes in initial plasma levels.	Increased cardiovascular output and blood flow may influence early drug distribution rather than absorption.	High adiposity alters plasma protein binding, potentially increasing free PTX concentrations. Low muscle mass (sarcopenia) may lead to higher peak plasma drug levels.	Exercise may help optimize early PTX distribution by enhancing vascular function. Patients with sarcopenia or high adiposity may experience altered plasma drug levels, requiring personalized monitoring.
**Distribution**	Greater volume of distribution (Vd) with higher lean mass.	Higher skeletal muscle mass enhances tissue perfusion and drug dispersion. Reduced adipose tissue prevents excessive retention in circulation.	Sarcopenia reduces Vd, leading to higher PTX plasma concentrations and increased toxicity risk. High adiposity alters PTX storage and plasma exposure.	Patients with low muscle mass may require lower PTX doses to prevent toxicity. Exercise that increases lean mass may improve PTX distribution and reduce side effects.
**Metabolism**	Enhanced CYP3A4 and CYP2C8 activity.	Exercise reduces systemic inflammation, indirectly improving hepatic metabolism via CYP enzymes.	Higher skeletal muscle mass is associated with increased hepatic enzyme activity, while adiposity-related inflammation suppresses metabolism.	Exercise may help optimize PTX metabolism by reducing inflammation. Obese patients may exhibit a slower metabolism and higher toxicity risk.
**Excretion**	Improved hepatic elimination and renal function.	Exercise reduces glomerular filtration transiently, but long-term exercise improves renal function.	Obesity and sarcopenia may alter hepatic elimination, affecting PTX elimination and toxicity risk.	Exercise may help regulate overall metabolic function, potentially lowering systemic toxicity in high-risk patients.

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
