# Peer review of "Synergy of Body Composition, Exercise Oncology, and Pharmacokinetics: A Narrative Review of Personalizing Paclitaxel Treatment for Breast Cancer"

_cancers, 2025, doi:10.3390/cancers17081271_

Round 1

Reviewer 1 Report

Comments and Suggestions for Authors

The manuscript by Adriaenssens and colleagues, entitled “Synergy of body Composition, Exercise Oncology and Pharmacokinetics: A Narrative Review in Personalizing Paclitaxel Treatment for Breast Cancer” provides a well written and stimulating discussion of paclitaxel and how its toxicity profile is impacted by more than a simple assessment of body surface area. The authors present evidence for incorporation of additional measures of skeletal muscle mass and adipose tissue into dosage guidelines in order to reduce chemo related toxicities. They present data on studies of body composition, chemo related toxicities, PK and exercise Oncology. Thee authors propose future research on integrating exercise interventions into patient care during and after cancer treatment in order to optimize paclitaxel outcomes. They also propose studies to refine dosing models that would better account for patient differences in drug metabolism.

This reviewer found the presentation to be well formulated and concise. The authors did a good job of incorporating the scientific literature of this area and presenting it well. They provide a well reasoned argument to support their ultimate conclusions and recommendations for improved dosing models and future research of incorporating physical activity into cancer treatments to improve outcomes and reduce toxicities. I believe the reviewers of the journal would find this manuscript to be of interest and I recommend publication of the manuscript.

Strengths: The manuscript was well organized and easy to read, with logical flow throughout.

Few weaknesses were identified. One area of concern, however, was found in section 2.2. This reviewer wonders if the inclusion of RDI in this manuscript is actually beneficial to the arguments being presented. Specifically, are lines 105 to 112 really necessary? In those lines, the authors describe relative dose intensity as an important measure that reflects dose delays or reductions in treatment that is based on the actual dose delivered compared to the planned dose intensity. However elsewhere in this manuscript the authors describe the inadequacies of the current models used to plan the dose intensity for a chemotherapy regimen and how it needs to be improved. This reviewer suggests that the main point of this section is really the last lines of the first paragraph of this section and the last paragraph, found in 119-124. Consider removing or re-writing lines 105-112.

A significant number of abbreviations are used in this manuscript. Consider including a summary of those somewhere in the document to improve readability. 

Author Response

We want to thank the reviewer for the thorough review work and suggestions to set our review even to a higher standard. Below, you can find a point-by-point response to each of the comments of the reviewer. The adaptations were made directly into the manuscript (uploaded).

The manuscript by Adriaenssens and colleagues, entitled “Synergy of body Composition, Exercise Oncology and Pharmacokinetics: A Narrative Review in Personalizing Paclitaxel Treatment for Breast Cancer” provides a well written and stimulating discussion of paclitaxel and how its toxicity profile is impacted by more than a simple assessment of body surface
area. The authors present evidence for incorporation of additional measures of skeletal muscle mass and adipose tissue into dosage guidelines in order to reduce chemo related toxicities. They present data on studies of body composition, chemo related toxicities, PK and
exercise Oncology. Thee authors propose future research on integrating
exercise interventions into patient care during and after cancer treatment in order to optimize paclitaxel outcomes. They also propose studies to refine dosing models that would better account for patient differences in drug metabolism.

This reviewer found the presentation to be well formulated and concise. The authors did a good job of incorporating the scientific literature of this area and presenting it well. They provide a well reasoned argument to support their ultimate conclusions and recommendations for improved
dosing models and future research of incorporating physical activity into cancer treatments to improve outcomes and reduce toxicities. I believe the reviewers of the journal would find this manuscript to be of interest and I recommend publication of the manuscript.

Strengths: The manuscript was well organized and easy to read, with logical flow throughout.

Few weaknesses were identified.

Comment 1: One area of concern, however, was found in section 2.2. This reviewer wonders if the inclusion of RDI in this manuscript is actually beneficial to the arguments being presented.
Specifically, are lines 105 to 112 really necessary? In those lines, the authors describe relative dose intensity as an important measure that reflects dose delays or reductions in treatment that is based on the actual dose delivered compared to the planned dose intensity. However
elsewhere in this manuscript the authors describe the inadequacies of the current models used to plan the dose intensity for a chemotherapy regimen and how it needs to be improved. This reviewer suggests that the main point of this section is really the last lines of the first
paragraph of this section and the last paragraph, found in 119-124. Consider removing or re-writing lines 105-112.

Response 1: We thank the reviewer for this thoughtful comment. We agree that the original phrasing may have inadvertently emphasized RDI as a fixed standard, despite our manuscript's overall aim to challenge the adequacy of current BSA-based dosing strategies. However, we consider RDI a clinically meaningful surrogate marker for treatment completion and therapeutic success, especially in the context of dose-limiting toxicities (DLTs). In this manuscript, we use RDI to highlight how patient-specific factors (e.g., body composition) influence treatment tolerability, often resulting in reduced RDI. Therefore, rather than remove this section, we have revised the text to better align with our central thesis: individualized strategies are needed to maintain therapeutic exposure while minimizing toxicity, and RDI remains a useful indicator of treatment tolerance and feasibility.

Section 2.2 is now reformulated as:

“Relative dose intensity (RDI), the ratio of the actual delivered dose intensity (mg/m² per week) to the planned dose intensity for a chemotherapy regimen, is a key measure that reflects dose delays or reductions during treatment  [20-22], however, its application must be interpreted in the context of evolving evidence highlighting the limitations of BSA-based dosing (see below in section 3). While Aan RDI below 85% is considered a clinically significant reduction from the standard therapy, these thresholds are typically derived from standard dosing models that do not consider individual patient charac-teristics, such as body composition. This decreased RDI is associated with worse out-comes, including reduced survival rates in advanced BC. This association has been demonstrated in both randomized clinical trials and retrospective observational studies [20, 23, 24]. Maintaining an adequate RDI is particularly important for BC patients un-dergoing PTX therapy, as timely administration of the planned dosage is crucial for achieving optimal prognosis. In most cancers, there is a plateau in the dose-response curve for cytotoxic chemotherapy. This means that increasing the dose towards an as-ymptotic threshold leads to increased toxicity without providing additional benefits to the antitumor effect [25]. Therefore, it is essential for patients undergoing chemotherapy to achieve a therapeutic response while minimizing toxicity, ensuring that their drug exposure remains at acceptable levels.”

Reviewer 2 Report

Comments and Suggestions for Authors

In the present manuscript author try to describe a novel concept. Overall manuscript looks good but I have a couple of minute comments before further proceeding. Such as 

  • Author needs to add to few more pictorials for great reader view .
  • The author needs to compare the pharmacokinetic aspect with large molecules.
  • Author need to provide further insights compare to current therapy of pharmacokinetic profile with different stage of tumor. 

Author Response

We want to thank the reviewer for the thorough review work and suggestions to set our review even to a higher standard. Below, you can find a point-by-point response to each of the comments of the reviewer. The adaptations were made directly into the manuscript (uploaded).

In the present manuscript author try to describe a novel concept. Overall manuscript looks good but I have a couple of minute comments before further proceeding. Such as :

Response:
For better guidance of the reader, we added a paragraph at the end of the Introduction about the overarching themes and what the reader could expect.

“In this narrative review, we explore how interindividual differences in body composition and physical activity levels may influence paclitaxel pharmacokinetics and the risk of chemotherapy-induced toxicities in breast cancer patients. We synthesize current evidence on the relationships between skeletal muscle mass, adipose tissue distribution, physical activity, and paclitaxel metabolism and clearance. Special attention is given to how unfavorable body composition—such as sarcopenia or sarcopenic obesity—can increase the risk of dose-limiting toxicities and how physical activity may mitigate these effects by positively influencing body composition and metabolic function (Figure 1). Finally, we highlight the emerging role of exercise oncology and its potential to contribute to more personalized, safe, and effective chemotherapy strategies.”

We have also created a new Figure (Figure 1, page 3) to clarify our concept. This introduces the reader to the concept of how PA & body composition influence chemotherapy-induced toxicities, before elaborating on the ADME principles in Figure 2.

Comment 2: The author needs to compare the pharmacokinetic aspect with large molecules.

Response:   

ADME processes of large molecules such as monoclonal antibodies (mAbs) are indeed different than those of small molecules such as paclitaxel. For example, clearance of mAbs primarily happens through receptor-mediated endocytosis, and not through hepatic or renal clearance. Additionally, due to their large size, mAbs are mainly confined to the intravascular space and extracellular fluid. Given these characteristics, the PK of mAbs are expected to scale less with differences in body size/composition (https://pubmed.ncbi.nlm.nih.gov/28754722/ ).

We therefore added the following statement to section 3.2.

“Conversely, the PK of large-molecule drugs, such as monoclonal antibodies (mAbs), are expected to be less influenced by body composition, as their elimination is receptor-mediated, and their distribution is primarily limited to the extracellular space.”

Comment 3: Author need to provide further insights compare to current therapy of pharmacokinetic profile with different stage of tumor. 

Response:   

Paclitaxel binds to tubulin and therefore accumulates to high concentrations in (cancer) cells. However, there is no evidence for target mediated drug disposition of paclitaxel at therapeutic concentrations. This implies that tumor size or abundance is not expected to alter the pharmacokinetic profile of paclitaxel (  It is possible that disease progression will be associated with physiogical characteristics that influence paclitaxel PK, such as decreased hepatic function. However, we are not aware of clinical evidence that supports adapting paclitaxel dose should be adapted according to the stage of the cancer. Current paclitaxel dosing is primarily adapted based on the occurrence of dose-limiting (hematologic) toxicities, with a maximum tolerated dose regardless of disease progression (https://www.ema.europa.eu/en/documents/product-information/abraxane-epar-product-information_en.pdf).  

Reviewer 3 Report

Comments and Suggestions for Authors

The manuscript provides a broad and relevant discussion on the interplay between body composition, exercise oncology, and paclitaxel pharmacokinetics. However, substantial revisions are necessary.

The manuscript should incorporate recent evidence on how body mass index (BMI) influences chemotherapy response. Specifically, it is highly recommended to cite:

Poggio F, et al. Efficacy of adjuvant chemotherapy schedules for breast cancer according to body mass index: results from the phase III GIM2 trial. ESMO Open. 2024 Aug;9(8):103650. doi: 10.1016/j.esmoop.2024.103650. Epub 2024 Aug 8. PMID: 39121814; PMCID: PMC11362642.

This study from the phase III GIM2 trial explores how adjuvant chemotherapy efficacy varies based on BMI, which directly relates to the discussion on body composition and pharmacokinetics.

The discussion on paclitaxel metabolism and clearance is too superficial. There should be a stronger connection between body composition and drug distribution/metabolism (e.g., increased clearance in obese patients). I suggest expanding the section on inter-patient variability in paclitaxel metabolism. It also important to discuss the role of obesity in altering drug distribution (e.g., higher volume of distribution in adipose-rich individuals).

The benefits of exercise in chemotherapy tolerance and efficacy need to be better substantiated with mechanistic data. I suggest discussing whether exercise-induced changes in body composition affect chemotherapy pharmacokinetics. It is also important to differentiate between the impact of aerobic vs. resistance exercise on body composition and paclitaxel metabolism. In addition, I suggest clarifying if exercise modifies chemotherapy-related toxicities or systemic drug exposure.

The manuscript suggests personalized chemotherapy dosing based on body composition but lacks practical implementation strategies. How should oncologists apply these findings in real-world practice? Discussion should be enriched by reporting whether BMI, skeletal muscle index, or other metrics should be used to tailor paclitaxel dosing.

Author Response

We want to thank the reviewer for the thorough review work and suggestions to set our review even to a higher standard. Below, you can find a point-by-point response to each of the comments of the reviewer. The adaptations were made directly into the manuscript (uploaded).

The manuscript provides a broad and relevant discussion on the interplay between body composition, exercise oncology, and paclitaxel pharmacokinetics. However, substantial revisions are necessary.

Comment 1: The manuscript should incorporate recent evidence on how body mass index (BMI) influences chemotherapy response. Specifically, it is highly recommended to cite:

Poggio F, et al. Efficacy of adjuvant chemotherapy schedules for breast cancer according to body mass index: results from the phase III GIM2 trial. ESMO Open. 2024 Aug;9(8):103650. doi: 10.1016/j.esmoop.2024.103650. Epub 2024 Aug 8. PMID: 39121814; PMCID: PMC11362642.

This study from the phase III GIM2 trial explores how adjuvant chemotherapy efficacy varies based on BMI, which directly relates to the discussion on body composition and pharmacokinetics.

Response:   

We appreciate the reviewer's suggestion to incorporate the important work by Poggio et al. However, after careful consideration, we maintain our focus on body composition analysis rather than BMI metrics for three key scientific reasons:

  1. Mechanistic relevance: Body composition parameters (muscle mass, visceral adipose tissue, etc.) directly influence paclitaxel pharmacokinetics as stated in the review section 3.b. like:

-            Adipose tissue distribution of lipophilic drugs

-            Systemic inflammation modulation affecting drug clearance

BMI lacks this physiological specificity, as shown by our Belgian colleague Desmedt*: BMI study that required additional body composition measures to explain survival differences.
*Desmedt C, Fornili M, Clatot F, Demicheli R, De Bortoli D, Di Leo A, Viale G, de Azambuja E, Crown J, Francis PA, Sotiriou C, Piccart M, Biganzoli E. Differential Benefit of Adjuvant Docetaxel-Based Chemotherapy in Patients With Early Breast Cancer According to Baseline Body Mass Index. J Clin Oncol. 2020 Sep 1;38(25):2883-2891. doi: 10.1200/JCO.19.01771. Epub 2020 Jul 2. PMID: 32614702.

  1. Scope alignment: The GIM2 trial focuses on survival outcomes (similar to Desmedt et al.) rather than pharmacokinetic parameters - an important distinction given our review's specific focus on drug exposure optimization rather than long-term clinical outcomes.
  2. Conceptual clarity: Including BMI-related survival data could confuse readers about our core thesis regarding direct body composition-pharmacokinetics relationships.

This selective approach maintains our review's tight focus while acknowledging BMI's role in broader clinical outcome studies.

Comment 2: The discussion on paclitaxel metabolism and clearance is too superficial. There should be a stronger connection between body composition and drug distribution/metabolism (e.g., increased clearance in obese patients). I suggest expanding the section on inter-patient variability in paclitaxel metabolism. It also important to discuss the role of obesity in altering drug distribution (e.g., higher volume of distribution in adipose-rich individuals).

Response:   

Due to the lipophilic nature of PTX, we acknowledge that the role of obesity is important in potential inter-individual variability in PTX pharmacology. As sarcopenia and other age-related factors influencing skeletal muscle mass were already addressed, we expanded the section focusing on obesity-related PK alterations.

“Similarly, obesity may alter drug distribution and affect PK by decreasing plasma concentrations of lipophilic drugs but increasing their storage in adipose tissue. For PTX, Sparreboom et al. found a 46.6% increase in Vd and a 20.4% increase in clearance in obese patients (BMI ≥30 kg/m2) compared to lean patients (BMI <25 kg/m2).Since PTX is strongly bound to plasma proteins (e.g. albumin), which are found to be reduced in obese patients [ref], this may lead to an increase in the free fraction of PTX, which in turn may increase its pharmacological effect, but also the risk of toxicity. PTX is metabolized in the liver by cytochrome P450 enzymes, primarily CYP3A4 and CYP2C8. Obese individuals may have decreased drug metabolism, as individuals with nonalcoholic fatty liver disease can have impaired hepatic drug-metabolizing enzyme activity, resulting in prolonged PTX exposure and increased risk of adverse reactions. Although PTX is primarily metabolized by the liver and excreted in the bile, there is some renal excretion. Therefore, impaired renal function, as observed in some obese patients [ref], can also adversely affect the elimination of this drug and its metabolites. In addition, women have been reported to experience greater toxicity than men at equivalent doses of PTX. This may be due to gender-related differences in body composition, such as women tending to have more body fat than men. The increased susceptibility of women to the adverse effects of PTX has also recently been linked to differences in the functioning of the immune system, although the underlying mechanism remains poorly understood. [refs] Therefore, it is important to be aware of interpatient variability and to consider differences in body composition when prescribing PTX.”

Comment 3: The benefits of exercise in chemotherapy tolerance and efficacy need to be better substantiated with mechanistic data. I suggest discussing whether exercise-induced changes in body composition affect chemotherapy pharmacokinetics.

Response:   

We thank the reviewer for this insightful comment. We agree that mechanistic insights are essential to understand how exercise-induced body composition changes may influence chemotherapy pharmacokinetics. In our current version, we already address this relationship at multiple points in the manuscript by integrating available evidence from clinical and preclinical studies.

For instance, in Section 5, we note that:

“Exercise-based interventions may mitigate these toxicities by maintaining SMM and improving metabolic function [122, 123].”

And in Section 6, we already discuss pharmacokinetic consequences of body composition variation:

“Patients with low SMM demonstrate higher PTX peak concentrations (C_max) despite similar AUC and clearance, due to a reduced volume of distribution (Vd) [131, 132].”

“Adipose tissue may also alter drug distribution and clearance, as PTX is a highly lipophilic drug and is sequestered in fat tissue, possibly reducing free drug availability and impairing elimination in obese individuals [134, 135].”

“Physical activity, by increasing lean mass and reducing visceral fat, may shift this profile toward improved metabolic clearance and reduced peak exposure…”

In light of the reviewer’s comment, we have added one additional sentence on preclinical studies related to body composition and exercise. For this sencente we refer to our response on comment 4 (below).

Response:   

We thank the reviewer for raising this important consideration. While current evidence specific to PTX metabolism remains limited, we acknowledge the relevance of differentiating between aerobic and resistance exercise in their respective effects on body composition and potentially on chemotherapy tolerance. We have now added a subsection in the manuscript (Section 5) to clarify these differences and highlight the clinical rationale for combining both exercise modalities in accordance with current guidelines. This includes a brief synthesis of known effects on lean mass and adiposity, and identifies current gaps in PTX-specific pharmacokinetic data. Until further studies emerge, clinical implementation should prioritize individualized programs aligned with ASCO guidelines promoting combined aerobic and resistance training.

In text, section 5 (correct references in the manuscript):

“While mechanistic differences exist between aerobic and resistance training, both modalities improve treatment tolerance by positively modulating body composition. Resistance training consistently increases lean mass (e.g., +8.3 kg vs. +2.7 kg with aerobic training) and reduces intermuscular fat infiltration, particularly beneficial for preserving skeletal muscle during chemotherapy [Ref]. In contrast, aerobic exercise yields more significant reductions in overall fat mass and percentage body fat (e.g., -1.8% vs. -0.9% for resistance training) [Ref].

Current ASCO guidelines recommend combining aerobic and resistance exercises to maximize clinical benefits across multiple outcomes [Ref]. Resistance training is es-pecially emphasized in patients at risk for sarcopenia or undergoing dose-dense chem-otherapy regimens. However, direct comparisons between aerobic and resistance exer-cise about PTX pharmacokinetics are currently lacking. Preclinical models suggest that combined exercise may enhance drug delivery through mechanisms such as vascular normalization, but these findings require clinical validation [Ref]. Therefore, until PTX-specific data become available, exercise prescriptions should be individualized based on patient capacity, safety, and tolerance, with a preference for multimodal pro-grams incorporating aerobic and resistance components.”

Comment 5: In addition, I suggest clarifying if exercise modifies chemotherapy-related toxicities or systemic drug exposure.

Response:  We agree that distinguishing between the effects of exercise on chemotherapy-related toxicities versus systemic drug exposure is important. In response, we have added a concise paragraph in Section 6 to clarify the mechanistic and clinical evidence supporting both effects. Specifically, we elaborate on how exercise-induced improvements in body composition may reduce paclitaxel toxicity by indirectly altering pharmacokinetic parameters such as Cmax and Tc > 0.05, which are known predictors of dose-limiting toxicities. We also note that while direct evidence remains limited, recent reviews and ongoing research are actively addressing this knowledge gap.

The beginning of section 6 now reads:

“Exercise may reduce chemotherapy-related toxicities both indirectly through improved physiological resilience and more directly via modulating systemic drug exposure. In the context of PTX, dose-limiting toxicities such as peripheral neuropathy have been linked to higher Cmax and prolonged exposure above the toxicity threshold (Tc > 0.05 μM) [27, 133]. Exercise-induced improvements in skeletal muscle mass and reductions in adiposity affect PTX pharmacokinetics, potentially lowering Cmax and shortening Tc > 0.05 due to changes in drug distribution and metabolism [104, 121, 133]. Additionally, exercise may enhance hepatic and renal perfusion, improve metabolic enzyme activity, and reduce systemic inflammation—factors that can promote drug clearance [104, 134, 135]. While direct clinical evidence of reduced systemic exposure through exercise remains limited, recent studies, such as the PABTOX study, have highlighted this mechanism as a promising hypothesis for reducing toxicities and improving chemotherapy adherence [92, 136, 137].”

Comment 6: e manuscript suggests personalized chemotherapy dosing based on body composition but lacks practical implementation strategies. How should oncologists apply these findings in real-world practice? Discussion should be enriched by reporting whether BMI, skeletal muscle index, or other metrics should be used to tailor paclitaxel dosing.

Response:   

We have now provided additional discussion on practical implementation strategies

Added to end of section 7:

“Clinically, low SMI may help identify high-risk patients who could benefit from extended infusion protocols, closer monitoring, or exercise interventions to improve muscle mass and treatment tolerance [27, 124, 125]. To facilitate this oncologists could use routinely acquired CT or PET-CT scans to assess e.g. SMI at the L3 level—a suggested proxy for total muscle mass that needs to be further explored [41, 61, 147]. Automated tools using artificial intelligence can extract SMI from scans without added burden [61, 148]. As imaging tools become standardized, integrating SMI or AT  into pharmacokinetic-informed dosing algorithms represents a promising step toward personalized chemotherapy. Beyond SMI alone, the ongoing PABTOX study (NCT06387901) mentioned above is developing a PK-PD model to integrate additional personalized metrics—such as SMM, AT, and PA—to improve early identification of patients at risk for dose-limiting toxicities and low relative dose intensity.”

Round 2

Reviewer 3 Report

Comments and Suggestions for Authors

The authors have addressed the comments.